# Reliability Assessment of NPP Safety Class Equipment Considering the Manufacturing Quality Assurance Process

Mohammad Khalaquzzaman [1,*], Seung Jun Lee [2] and Muhammed Mufazzal Hossen [1]

1   Nuclear Power and Energy Division, Bangladesh Atomic Energy Commission, Agargaon,
    Dhaka 1207, Bangladesh; mufa50du@yahoo.com
2   Department of Nuclear Engineering, Ulsan National Institute of Science and Technology, 50 UNIST-gil,
    Ulsan 44919, Republic of Korea; sjlee420@unist.ac.kr
*   Correspondence: kzaman@rooppurnpp.gov.bd

**Abstract:** Quality and safety are intensely related and go hand in hand. Quality of the safety-grade equipment is very important for the safety of a nuclear power plant (NPP) and achieving production goals. During manufacturing of plant components or equipment, deviation from the design might occur at different stages of manufacturing for various reasons, such as a lack of skilled manpower, deviation of materials, human errors, malfunction of equipment, violation of manufacturing procedure, etc. These deviations can be assessed cautiously and taken into consideration in the final safety analysis report (FSAR) before issuing an operating license. In this paper, we propose a Bayesian belief network for quality assessment of safety class equipment of NPPs with a few examples. The proposed procedure is a holistic approach for estimation of equipment failure probability considering manufacturing deviations and errors. Case studies for safety-class dry transformers and reactor pressurizers employing the proposed method are also presented in this article. This study provides insights for probabilistic safety assessment engineers and nuclear plant regulators for improved assessment of NPP safety.

**Keywords:** reliability; quality factor; safety; Bayesian belief network

## 1. Introduction

Equipment performance and safety depend on product quality. Quality assurance (QA) programs for the design and manufacture of equipment play a substantial role in product quality. Poor design, production, and testing facilities degrade equipment quality [1]. Quality assurance in the nuclear industry is the most important task for protecting people and the environment [2]. It is worth mentioning that good quality assurance means a stringent quality assurance system with a detailed quality program complying with international standards and defined manufacturing procedures. On the other hand, poor quality assurance indicates a lack of a standard quality management system. Probabilistic safety assessment (PSA) of a nuclear power plant (NPP) is a regulatory requirement for NPP construction licensing. PSA provides vital input to the regulators and operating organizations of NPPs regarding the reliability of plant equipment and safety systems. Presently, preliminary safety analysis is performed prior to issuing an NPP construction permit, and the final safety analysis is performed after commissioning through qualitative and quantitative assessment considering deviations in equipment manufacturing and construction of the plant. However, reliability assessment of equipment can be performed through detailed evaluation of the manufacturing stages and by quantifying the actual deviations in the manufacturing process to achieve a better assessment result.

Reliability analyses of manufacturing work have been conducted over the years employing various techniques, including failure mode and maintenance analysis (FMMA), failure mode and effect analysis (FMEA), failure mode effect and criticality analysis (FMECA), and Six Sigma [3–5]. FMEA is usually used to assess product reliability, which requires

identification of failure modes of a specific component or system, as well as frequency of failure and potential causes, based on which the risk priority number is calculated [6]. FMEA has also been used for reliability analysis in process industries, and some research regarding how to combine a Bayesian belief network (BBN) and FMEA to predict error probability in manufacturing processes has been carried out [7].

In this paper, we propose a BBN model to measure equipment quality, considering an assessment of the quality control activities performed in an equipment manufacturing plant. The BBN approach has been extensively employed over the years in different fields. The model has been employed for the reliability assessment of vehicles, electrical equipment, safety-critical software, etc. [8–13]. BBN can be used as a promising tool for quality assessment of nuclear safety-grade products, for which continuous safety assessment is necessary.

In PSA, the reliability of safety-critical equipment is determined by expert judgments if field and laboratory test data are unavailable or insufficient. Since the reliability of equipment is directly influenced by the quality assurance programs executed during equipment manufacturing, estimating the quality factor through an in-depth assessment of manufacturing quality assurance activities leads to better PSA outcomes. The aim of this research is to develop a methodology for equipment reliability estimation, which is presented in this article.

In this study, we estimate equipment quality factors and identify the necessity of considering the manufacturer's quality assurance activities. We discuss applications of the BBN methodology for quality assessment with two case studies, i.e., a case of a dry transformer and a case of a reactor pressurizer of a nuclear power plant. In nuclear power plants, many safety-class dry transformers are used in reactor building to feed power to safety-class electromechanical and electronic devices. As an example of mechanical equipment in this study, we selected a reactor pressurizer, which is a safety class-1 equipment that works under the reactor pressure boundary. On the other hand, we selected a safety class-2 dry transformer as an example of electrical equipment and a pressurizer as an example of mechanical equipment. The procedure can be similarly applied to nuclear fuel assemblies, reactor pressure vessels, steam generators, sensors, and reactor protection systems.

## 2. Modeling Approach for Manufacturing Quality Control Processes

Manufacturing of NPP safety-grade equipment is accomplished by targeting the nuclear safety requirements. Criteria for safety-grade equipment qualifications are described in different publications [14–16]. Hernandez discussed the quality control procedure for manufacturing nuclear safety-grade equipment [17]. A basic scheme of quality assurance (QA) activities performed in the manufacturer environment involving internal and external inspection organizations is presented in Figure 1. The QA procedure is usually designed by the QA team of the manufacturer. The volume and coverage of activities depends on the safety class and the importance of the equipment in a nuclear power plant based on the graded approach philosophy.

The QC team of a manufacturing plant is responsible for developing and executing a quality plan. The QC inspector performs the quality inspection schedule for the manufacturing stages. However, the owner and regulatory organization have the scope to review the quality plan and make choices regarding participation in quality inspections. They may decide to participate in the inspection stage as a witness in a graded approach to verify compliance with design and normative requirements. Depending on the type of equipment, the most common inspection activities under the different manufacturing stages include a readiness check by manufacturers; material control inspection; check of welding procedures, fit-up, and welder qualifications; non-destructive testing of materials and welds; machining checks; inspection of forging and heat treatment; and acceptance tests (e.g., hydraulic test, load test, etc.). Manufacturing stage inspection points are named hold points (HP), witness points (WP), or witness with report (WPR). In the case of HP, the

production process remains suspended until quality inspection is completed by inspectors. HP is considered the most critical stage of manufacturing.

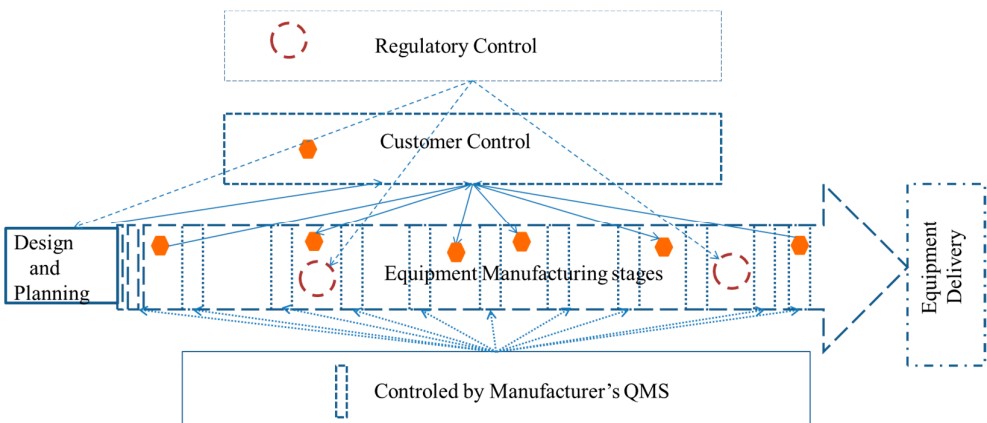

**Figure 1.** Basic structure of a QA system of typical NPP safety-class equipment.

We developed a methodology for the evaluation of equipment quality through the assessment of the QA activities shown in the quality control scheme (Figure 1). In this regard, a quality factor is estimated, which is assumed to be unity for perfect manufacturing. The quality factor is estimated by evaluating QA activities employing our model. Finally, the reliability of equipment is rated based on the quality factor. Our modeling approach is shown in Figure 2. The evaluation of the quality of an example product by a BBN is shown in the next section.

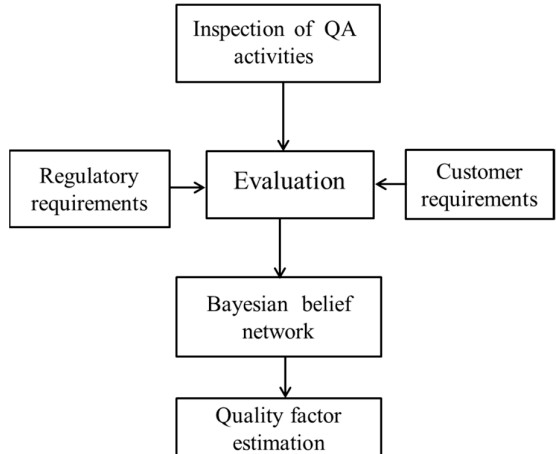

**Figure 2.** Modeling approach applied in this study.

## 3. Modeling with BBN

A BBN consists of graphical representations of nodes in which the probabilistic dependency between the interconnected nodes is set. A node represents a variable, and a linking arc between two nodes represents the causal relationship between variables [18]. In a BBN, all of the nodes related to the attributes for a particular task are connected to a child node. The direction of the arrows represents the impact of the parent nodes on the child nodes. When a single child node has multiple parent nodes, a probability table (PT) is required to describe the influence of parent nodes on the child node. The complexity of the node probability increases with an increase in the number of parent nodes. The process of creating an NPT is described in the manual of BBN tools (AgenaRisk, Netica).

In a BBN, a connection is considered divergent when a parent node possesses multiple child nodes and influences are passed to all of the child nodes belonging to the parent

node [18]. In a diverging connection, the state of the parent node can be inferred if the states of the child nodes are known or vice versa. To explain the calculation process of a diverging model, initially, a very simple model consisting of two nodes (a parent (*Y*) and a child (*X*)) is considered. If *X is observed as "high"*, the probability of *Y being high is estimated by applying* Bayes' rule as follows:

$$P(Y = high | X = high) = \frac{P(X = high | Y = high)P(Y = high)}{P(X = high)} \quad (1)$$

NPTs contain probability information based on belief relationships between parent and child nodes in a model.

The BBN method is also applied to evaluate software development lifecycle activities to estimate unrevealed faults in software. The technique has also been applied in some non-nuclear applications [19].

The US NRC developed a procedure for BBN modeling which was proposed for reliability assessment of an NPP digital protection system. The major steps in the development a BBN model for equipment quality are summarized as follows: familiarization with the equipment manufacturing process and international standards applicable to NPP safety-critical equipment manufacturing, construction of a Bayesian graph, creation of a probability table, preparation of input data and expert elicitation, and performing analyses and generating results [9,20].

Fenton et al. discussed the principle of ranked nodes considering a truncated normal (TNormal) distribution [11]. The probability distribution of a three-state ranked node is shown in Figure 3.

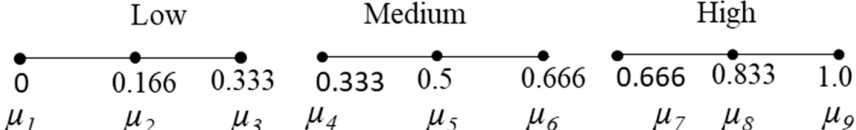

**Figure 3.** Probability distribution of a three-state ranked node.

The basic BBN estimation procedure for the three-state ranked node considering TNormal distribution is as follows:

If $P(B = low | A = low)$ is the average of three sample points in state High: $(\mu_1, \mu_2, \mu_3)$.

$$P(B = Very\ Low | A = Very\ Low) = \frac{\sum_{k=1}^{3} \int_{x=0}^{1/3} TNormal\left(x, \mu_k, \sigma^2, 0, 1\right)}{3} \quad (2)$$

where $\mu$ is the mean, and sigma is the standard deviation. Similarly, the probability ($P(B = low)$) when the node *A* is low, medium, and high is estimated.

To estimate a quality factor employing a BBN, a model is constructed and quantification is performed based on the inputs gathered from manufacturing quality control certificates and test and inspection reports. A qualitative assessment of each manufacturing task is performed against a set of regulatory and normative requirements (ASME, IEC, GOST, etc.). Over the decades, BBN models have been applied in software reliability modeling [21–23]. The progress and recognition of the BBN technique in probabilistic risk assessment was discussed in [24]. The NKS report on guidelines for reliability analysis included BBN modeling [25].

## 4. Case Studies

### 4.1. Case 1: Dry-Type Transformer

Transformers are essential elements of the power system of an NPP, and hundreds of transformers are required for a large NPP. A large number of dry-type transformers are used in nuclear power stations. Usually, dry transformers are chosen for the hazardous area to reduce maintenance and the probability of fire incidents [26].

Operational records show that the failures of dry transformers were mostly detected during operation of the transformer. Failures during operations were identified to account for as many as 84% of the total events, with the rest of the failures were identified during maintenance and in-service inspections. The most significant mode of failure of this type of transformer was winding failures, and the failure of cooling fans caused winding failure [27,28]. A case study for quantitative evaluation of the quality of a dry transformer using BBN is presented. We developed a model with inputs based on the quality control steps of a dry transformer presented in the following subsections.

### 4.1.1. Formulating Attributes of a BBN for a Dry Transformer

Quality control of transformer winding, core fitting, resin insulation encapsulation, and assembly of the accessories play vital roles in the performance and length of service life of transformers. A stringent quality assurance program for design, fabrication, assembly, and testing can considerably improve the quality of a transformer. We reviewed several quality plans for dry-type transformer manufacturing. The quality control activities and regulatory requirements are shown in Table 1. The BBN model presented in Section 4.1.2 was developed by analyzing QC activities and regulatory requirements.

**Table 1.** QC activities for dry-type transformer manufacturing.

| Name of QC Activities | Normative Requirements |
|---|---|
| Inspection of readiness to commence manufacturing | The manufacturer shall have a quality assurance program; a license for HV equipment manufacturing; skilled manpower (supervisor, electrician, mechanics, and welders); environment, health, and safety management certificates; standard approved procedures for manufacturing, testing, preservation, and packing; accredited laboratories and test facilities; detailed design and working documentation; an inspection and test plan; and calibration certificates of measuring equipment. |
| Control of materials and off-the-shelf equipment | Visual inspection of materials, check of marking, procurement certificates, and quality conformation certificates. |
| Inspection of HV winding before casting | Check of production procedure, completeness of windings, welders' qualification, soldering and welding execution procedure, certificates of welding and soldering materials, fit-up of welding, assembly and disassembly of mandrels, and heat treatment of windings. |
| Manufacturing inspection of LV winding | Check of production of LV winding, completion of windings, soldering and welding, assembly, disassembly of mandrels, and heat treatment of LV windings. |
| Control of the manufacture of the magnetic core | Check of production of the transformer magnetic core and visual measuring inspection of cross cutting of electrical steel as per drawing. |
| Control casing manufacture and assembly of the casing before coating | Visual check and dimensional measuring of assembly units and casing parts in accordance with the approved casing specification and drawing. |
| Check of painting preparation | Visual check of assembly units and casing parts in accordance with the casing specification. |
| Quality control of paint and varnish coating | Check of the shelf life of paint and visual inspection and measurement of the coatings on the assembly units and casing parts in accordance with the specification. |
| Inspection of the assembly of the frame with core and windings | Check of manufacturing skeletons with windings, coatings, and cores with winding assemblies as per approved procedure. |

| Name of QC Activities | Normative Requirements |
|---|---|
| Control of the assembly of the transformer and installation in a protective casing | Visual check of the assembly of the active part and final assembly and in accordance with the assembly drawing. |
| Type and routine test | Check of the execution of type tests and routine tests as per approved plan. |
| Preservation, packaging of the transformer, marking, and check of shipping documents | Check of completion, preservation, and packaging of goods as per standard dimensional drawing and check of documents. |
| Acceptance inspection | Check of quality control reporting documentation, completeness of products, and technical and shipping documentation. Inspection of marking and paint thickness and preservation in accordance with the requirements. Visual inspection of packages. |

### 4.1.2. Development of the Network

BBN subnets for the different quality control activities mentioned in Table 1 are shown in Figures 4–7. Each of the nodes is considered to have five ranks of quality: *very low*, *low*, *medium*, *high*, and *very high*. The probability of parent nodes is inferred based on the NPT and the status of indicator nodes.

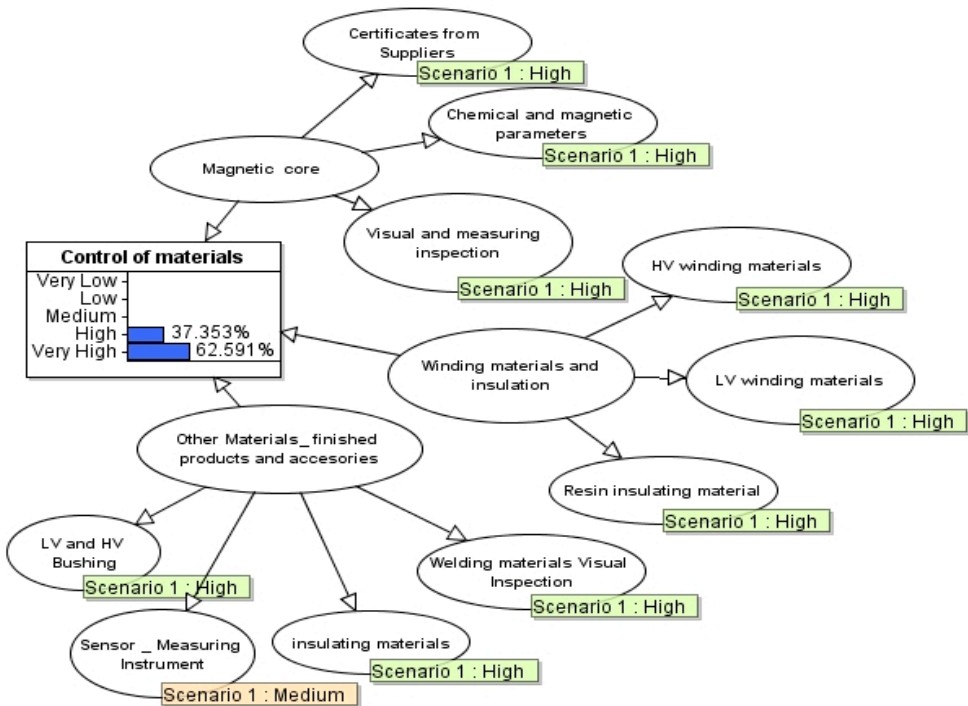

**Figure 4.** Subnetwork for assessment of material control.

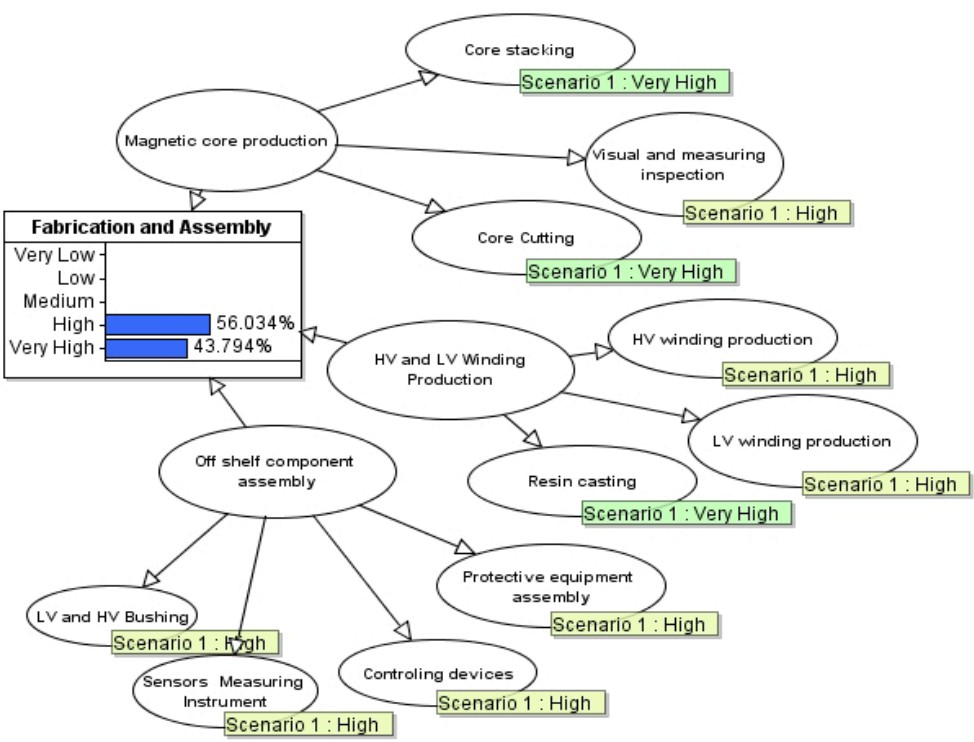

**Figure 5.** Subnetwork for assessment of production quality.

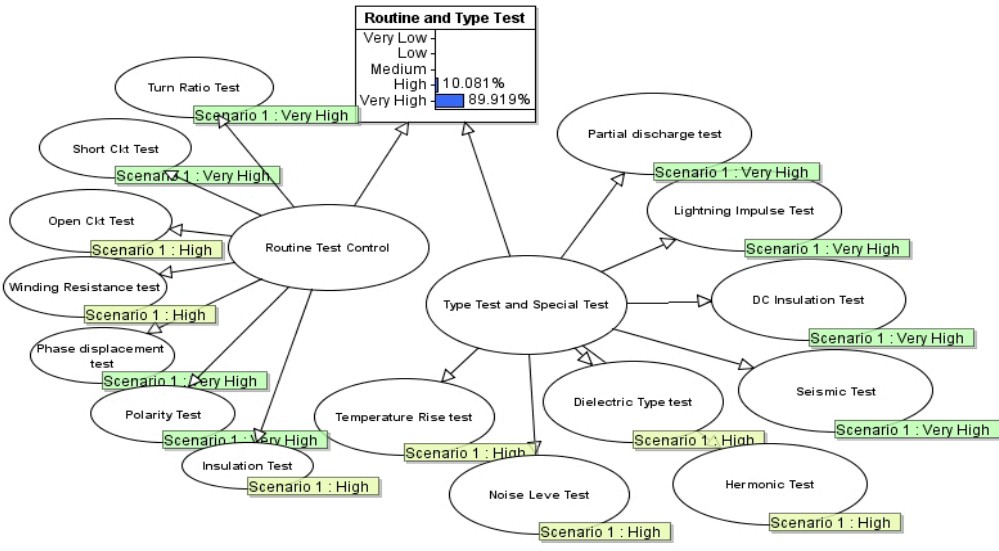

**Figure 6.** Subnetwork for assessment of electrical tests.

Combining the BBN subnetwork for a dry transformer, the overall quality assessment result was obtained from the network shown in Figure 7. The model showed that the probability of manufacturing quality being "*very high*" was 71.34%, and the probability of "*high*" manufacturing quality was 28.64%, corresponding to an overall quality factor of 0.942 considering the worth factor of each rank level (i.e., 0.2 for *very low*, 0.4 for *low*, 0.6 for *medium*, 0.8 for *high*, and 1.0 for *very high*).

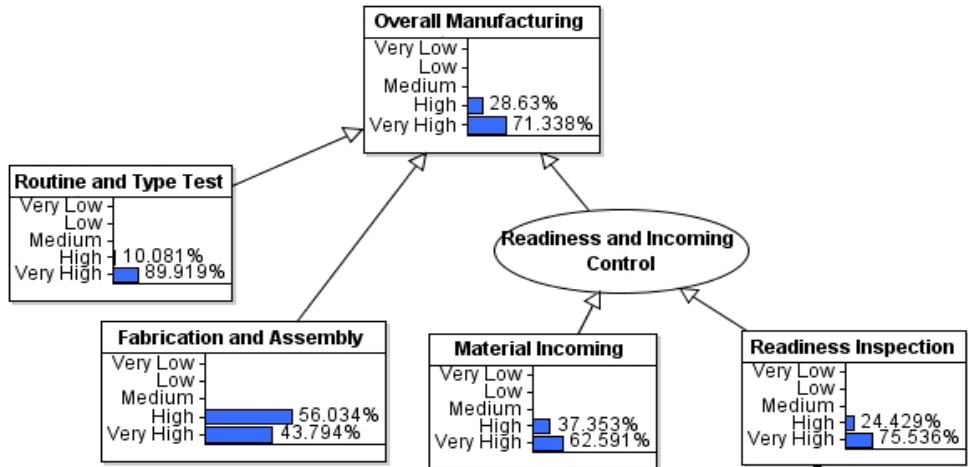

**Figure 7.** Network combining all QC activities.

4.1.3. Prediction of Transformer Failure Probability

Exponential distribution with a constant random failure rate is widely used in the reliability analysis of electrical and semiconductor devices [29,30]. If equipment wear-out exists, a lognormal distribution is considered for reliability estimation. In the case of mechanical devices, the effects of corrosion and wear reduce the mean time between failures. However, extensive quality control during manufacturing significantly extends the useful life of such devices. The probability of hardware failure for electronic components of a system ($q_{rd}(t)$ for $0 < t < \tau$) is expressed as follows:

$$q_{rd}(t) = 1 - e^{-\lambda t} \tag{3}$$

where lambda ($\lambda$) represents a constant random failure rate. The magnitude of the $\lambda$ of equipment is significantly inclined to the operating environment, as well as thermal stress, design, and manufacturing quality. Thus, considering the causes of the degradation, the value of $\lambda$ can be revised to a lower value from the ideal level. $\lambda^m$ represents a modified random failure rate, taking manufacturer QA and QC programs into consideration, and can be estimated as $\lambda/q$. Hence, "$q$" represents a quality factor; the highest value is 1, corresponding to excellent QA during manufacturing. The value of $q$ was estimated by applying the BBN model, and component failure probability was estimated.

The quality factor predicted by the BBN model shown in Section 3 was applied to estimate the reliability of a transformer, with results presented in this section. The failure probability of a transformer is calculated based on the exponential failure distribution. The failure probability of a transformer was estimated as a base value for the nominal constant random failure rate ($\lambda = 0.8/10^6$ h). Taking into account the quality factor predicted by the BBN, the predicted failure rate was re-estimated as $\lambda^m = \lambda/q$.

Figure 8 shows the distribution of failure probability of a dry transformer over the service life of 20 years, in which the reliability of the periodically maintainable components (calibration of thermometers and cooling fans, tightness of connections and frames, etc.) and non-maintainable components (e.g., cast resin windings) were considered. The periodic maintenance scope of a dry transformer is smaller than that of an oil-immersed transformer. However, the periodically maintainable components are taken into account for a more pragmatic estimation. The failure probability considering the periodic and non-periodically maintainable components can be estimated as follows.

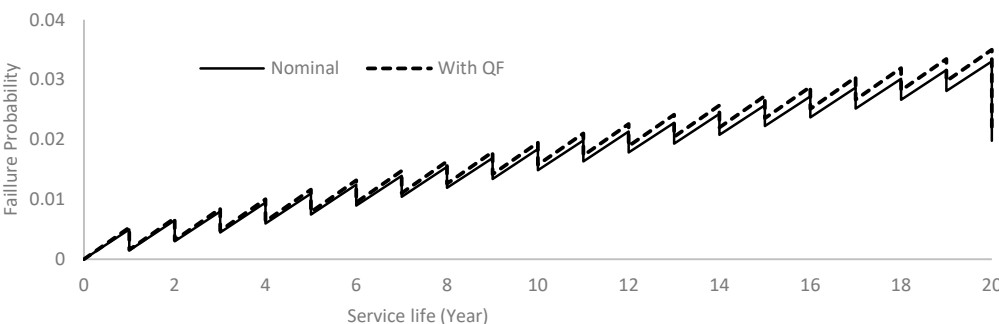

**Figure 8.** Failure probability of the transformer for the predicted failure rate ($\lambda^m$).

Assume that the failure probabilities of the periodically maintainable portion and non-maintainable portion are $q_p$ and $q_{np}$, respectively. Then, the combined failure probability ($q_t$) can be stated as follows:

$$q_t = q_p + q_{np} - q_p q_{np} \qquad (4)$$

Considering the manufacturing quality factor, the failure probability ($q_t^m$) was re-estimated using Equation (5):

$$q_t^m = q_p^m + q_{np}^m - q_p^m q_{np}^m \qquad (5)$$

*4.2. Case 2: Quality Assessment of a Reactor Pressurizer*

A basic BBN model to assess the manufacturing quality of a reactor pressurizer is presented in this section. The attributes of the model are based on the major quality control activities performed during manufacturing of the pressurizer components. The activities listed in Tables 2–4 are assessed against the normative requirements and customer requirements (if any). The evidence and records of work are considered as verifiable indicators.

**Table 2.** QC activities prior to production of pressurizer components.

| Name of QC Activities | Normative Requirements |
|---|---|
| Check of manufacturer preparedness | Manufacturer shall possess a quality assurance program, state license for design and manufacturing of nuclear grade equipment, sufficient qualified technical staff, certificate for protection of environment, health and safety, QMS (e.g., ISO 9000) certificate, standard, procedures (manufacturing, test procedures, preservation, packing), accredited laboratories and test facilities. Approved detailed design documentation of the items intended to manufacture, inspection and test plan, certificates for metrological equipment. |
| Inspection of non-destructive test materials | Check of materials in accordance with the standard. |
| Inspection of base materials and semifinished products | Check of availability, compliance of certificates, marking, materials as per drawing and normative requirements, visual and measuring inspection. |
| NDT of base metal and semifinished products | Necessary ultrasonic test according to the normative requirements. |
| Check of welding and cladding materials | Visual inspection of welding materials, check of chemical composition, mechanical properties, inter-crystalline corrosion resistance, and ferrite phase content on trial weld. |

**Table 3.** QC activities related to fabrication of pressurizer components.

| Name of QC Activities | Normative Requirements |
| --- | --- |
| Inspection of forgings | Marking, compliance of certificates with the requirements of design documentation, and visual inspection |
| Check of welding, cladding procedure, specifications, and welder performance qualification | Check of WPS and WPQ |
| Check of welding fit-up | Inspection of marking and visual and measuring inspections of preparation and fit-up for welding as per requirements |
| Check of parts, welding process, and welds | Inspection of welding process and visual and measuring inspections of welds as per requirements |
| NDT of welds | Penetrant test or radiographic test of part welds as per requirement |
| Check of machining of parts | Inspection of marking and visual and measuring post machining inspections as per drawing requirements |
| Check of heat treatment | Inspection of heat treatment process |

**Table 4.** QC activities related to acceptance test of pressurizer components.

| Name of QC Activities | Normative Requirements |
| --- | --- |
| Check of load test | Visual inspection of test process and inspection of welds after release of test load as per technical requirements |
| Hydrostatic tests | Check of water quality for hydro testing. Visual and measuring inspections of the pressurizer during hydro testing as per the test program. NDT of welds after hydraulic test. |
| Check of pressurizer inner-surface cleanness | Visual inspection of pressurizer inner-surface cleanness after hydraulic tests as per technical requirements of the approved procedure |
| Trial assembly with thermal electric heater | Trial assembly of the pressurizer with a tubular electric heater unit or a simulation of a tubular electric heater unit as per technical requirements of the drawing |
| Measuring inspection of pressurizer | Inspection of shells of after hydro tests as per technical requirements |
| Acceptance inspection | Check of reporting documentation on quality control, completeness of products, and technical and shipping documentation; check of marking; visual inspection of finished item; visual inspection of surfaces after painting and preservation; and packing as per requirement |

### 4.2.1. Manufacturer Preparedness

Quality control inspectors assess manufacturer preparedness before starting production to confirm that manufacturer preparation is sufficient to launch the production process of reactor pressurizer components. Major requirements for passing the readiness control point are the availability of the required state license for manufacturing safety-grade equipment, a quality assurance program, a detailed design, working documentation, inspection, test plans, qualified technical staff for welding, a non-destructive test, etc. In addition, the manufacturer must have environmental, health, and safety management certificates; standard approved procedures for manufacturing, testing, preservation, and packing; accredited laboratories and test facilities; and calibration certificates for measuring equipment. A BBN model for assessment of the readiness of manufacturers is shown in Figure 9.

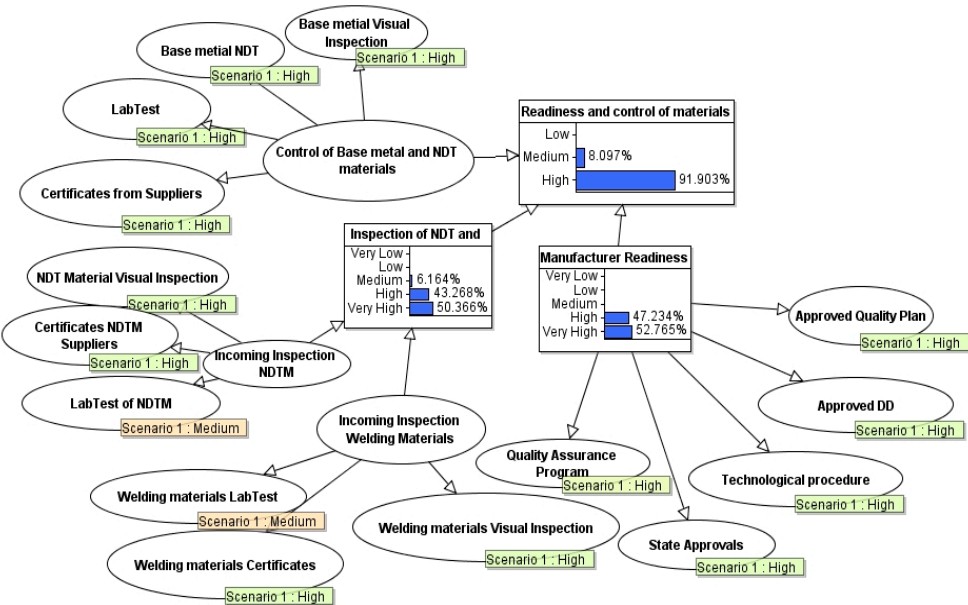

**Figure 9.** Subnetwork for readiness and material control activities.

### 4.2.2. Control of Materials

Control of base metals, welding materials, and non-destructive testing materials is performed as per regulatory and design requirements by checking the supplier certificates. A second test of the materials to confirm the chemical composition of the materials as per requirements can be performed at the laboratory of the manufacturer's plant. Visual inspection of materials is carried out to check the material surface condition, geometry, and quantity; stamps or markings of the producers on the materials are also examined. A test for base metal chemical composition includes verification that elemental contents—C, Mn, Si, P, S, Ni, Cr, Cu, Mo, V, W, Ti, and Fe—are within the acceptable ranges. A chemical analysis of welding wire is required to check the levels of C, Si, Mn, P, S, Cr, Ni, and Mo. Tests of the mechanical properties—hardness, yield strength, ultimate strength, and elongation—must be checked during incoming control of the welding materials.

### 4.2.3. Control of Welding

Welding assessment is performed by checking pre-welding activities and welding execution, as well as by post-welding tests, as per requirements. Pre-welding activities include the availability of the approved procedures, fit-up, and checking of welding materials (e.g., electrodes, wire, flux), as well as the qualifications of welders. Activities related to welding fit-up include visual inspection of surface cleanliness, alignment, and weld edges, as well as gap measurements, as per normative requirements.

### 4.2.4. Hydraulic Testing

Hydraulic tests are carried out to ensure the strength and integrity of equipment as per requirements. Test preparation includes ensuring the availability of valid documents for all of the measuring instruments to ensure their proper calibration. Safety of people and equipment must be ensured prior to the start of the test. Water quality is ensured by checking the laboratory certificate for the parameters, e.g., PH value at a certain temperature and contents of salts and chlorides. Measuring instruments are set as per the test diagram. The water temperature is increased at a prescribed rate (e.g., 30 °C/h for the reactor pressurizer). During pressurization, the buildup pressure rate (i.e., 1 MPa/min) and hold pressure are set as per technical requirements and must be in line with the hydraulic test pressure curve. Visual inspection is required at the prescribed reduced pressure to ensure the integrity and non-deformation of the vessel.

### 4.2.5. Delineating Network for Reactor Pressurizer Manufacturing QC Activities

BBN subnets using the quality control activities for a reactor pressurizer listed in Tables 2–4 are presented in Figures 9–11. Each of the nodes in this model are also considered to have five ranks of quality, and the probability of parent nodes is inferred based on the NPT and the status of indicator nodes.

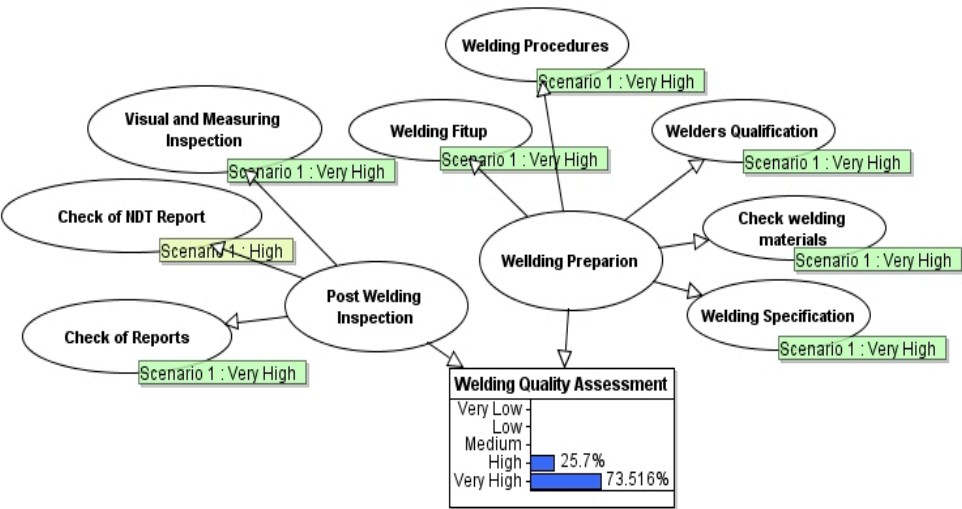

**Figure 10.** Sub network for welding quality assessment.

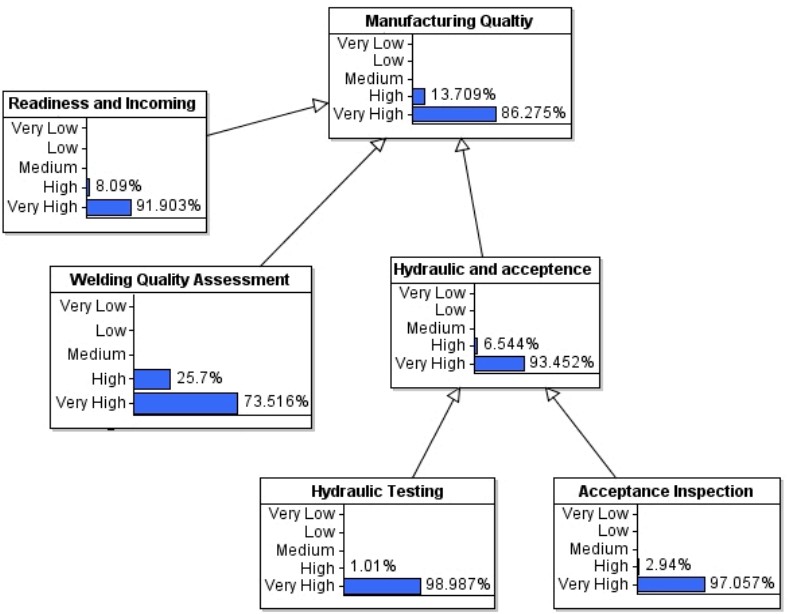

**Figure 11.** Subnetwork for overall quality assessment for the reactor pressurizer.

### 4.2.6. Estimation of Quality Factors

The overall quality assessment of the pressurizer components by the BBN model is shown in Figure 11. The model shows that the probability of *very high* quality of manufacturing is 86%, and the probability of *high* manufacturing quality is 14%, corresponding to an overall quality factor of 0.97 considering the worth factor of each rank level (i.e., 0.2 for *very low*, 0.4 for *low*, 0.6 for *medium*, 0.8 for *high*, and 1.0 for *very high*).

## 5. Result and Discussion

A BBN approach for quality assessment of equipment through the assessment of the QA and QC activities executed at the manufacturing plant was proposed in the previous

section. The model was employed for the case of an electric power transformer and reactor pressurizer component.

In this study, the quality factor was assumed to be unity for excellent manufacturing quality, which fulfills all the requirements. The quality factor estimated by the model was less than unity due to the deviation of the QC activities from the ideal conditions. The quality assessment of dry transformer by the BBN model is shown in Figure 7. The model estimated the probability of a *very high* quality of the product to be 71.34% and the probability of a *high* product quality to be 28.64%. The overall quality factor is estimated by taking into account the worth factor of each rank (such as 0.2 for *very low*, 0.4 for *low*, 0.6 for *medium*, 0.8 for *high*, and 1.0 for *very high*). The model estimated a quality factor of 0.94, which was used to predict the failure probability of the transformer. The influence of the poor-quality factors is shown in Figure 8. Poor quality control leads to early failure of the device and eventually decreases the design life of the equipment. The rate of increase in failure has an impact on the reliability of non-periodically and periodically maintainable components. Moreover, the result of the quality assessment provides a basis for justification of the storage of spare parts and the maintenance schedule.

The manufacturing process of a dry transformer is more complicated than that of an oil-emerged transformer. The key components (e.g., transformer windings and insulation) are not periodically maintainable. During an annual periodic inspection, dust and greasy layers deposited on the surface are cleaned to maintain heat removal capacity at the required level. In the case of periodically maintainable components, the quality factor estimated by the model helps to determine the frequency of periodic maintenance.

The second model was considered for an NPP pressurizer vessel, as presented in Figures 9–11. The NPP reactor pressurizer is safety-class equipment. Failure of the pressurizer (e.g., leakage; fracture of vessels, joints, valves, etc.) introduces LOCAs (loss of coolant accidents) or can be an initiating event for small or medium-size LOCAs. The quality factor can be used to modify the pressurizer failure rate. Quality factors can be assessed in a similar manner as shown for the dry transformer. The model shows that the probability of a *very high* quality of manufacturing is about 86%, and the quality of *high* manufacturing is about 14%, which corresponds to an overall quality factor of 0.97 considering the worth factor of each rank. Corrosion and wear-out of mechanical devices cause random failures during the operational phase. Manufacturing quality assessment helps to determine the level of attention required for equipment maintainability and slowdown of corrosion and in determining periodic maintenance frequency. In the case of unavailability of sufficient data, sensitivity analyses can be performed.

In our modeling, the attributes belonging to the terminal nodes are assumed to have an equal impact on the parent nodes. However, in reality, consideration of the weighting factors provides better results, since different quality control activities have different levels of impacts on product quality. For instance, verification of base materials should have a greater impact compared to the verification of non-destructive test materials.

Evidently, inspection at the manufacturing stage has a positive impact on the quality of components. Higher inspection frequency provides more opportunities for correction of errors, unplanned or inadvertent use of substandard materials, or change of class of materials in the case of scarcity of materials, as well as opportunities to save time. Likewise, inspection can prevent skipping of redundant tests set in the quality plan. Preparation of a stringent quality plan with a larger scope of testing and stage inspections can improve the quality of equipment.

## 6. Conclusions

A new approach for reliability estimation of safety-critical equipment of an NPP was proposed in this study, and BBN methodology was applied for the assessment of manufacturing QA activities. A BBN model was developed to measure equipment quality for nuclear safety-grade products of an NPP, which was applied to reliability estimation of a nuclear reactor pressurizer and dry-type transformer.

The graded approach philosophy in internal and external inspection is applied to NPP equipment manufacturing. QA activities are executed as per the approved quality plan, and the QC inspector performs quality inspection in various manufacturing stages with the participation of manufacturers, owners, and regulators. The records of the execution of QA activities, test certificates, etc., are primary indicators and are used as input to the model.

The notable contribution of this study is the development of a BBN model to measure equipment quality for nuclear safety-grade equipment of an NPP. This BBN model can be applied as a means of assessing manufacturing quality control activities and provide inputs to the PSA required in nuclear power industries, as well as safety-critical industries. The successful application of reliability estimation of a nuclear reactor pressurizer and dry-type transformer for an NPP demonstrates the applicability of the model proposed in this study. Although model developed in this study was applied to two specific pieces of equipment, the methodology is applicable to any safety-class or non-safety-class equipment.

Moreover, the proposed approach of reliability estimation of safety-critical equipment considering the manufacturing quality assurance process can be applied to improve the equipment manufacturing process and determine preventive maintenance policies and requirements for spare parts. In traditional PSA, input data for the basic events are determined through expert judgment in cases in which field data are lacking. The approach proposed in this study can be applied to the estimation of equipment reliability when equipment failure data are insufficient or unavailable.

**Author Contributions:** Conceptualization, M.K.; methodology, M.K. and S.J.L.; software, M.K.; validation, S.J.L., M.K. and M.M.H.; formal analysis, M.K.; investigation, S.J.L.; data curation, M.M.H.; writing—review and editing, M.K., S.J.L. and M.M.H.; supervision: S.J.L. All authors have read and agreed to the published version of the manuscript.

**Funding:** This research received no external funding.

**Data Availability Statement:** No data sharing is applicable to this article.

**Acknowledgments:** We acknowledge the Agenarisk authority for the trial version of the BBN tool that was applied for modeling.

**Conflicts of Interest:** The authors declare no conflict of interest.

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
