# Peer review of "Reliability Assessment of NPP Safety Class Equipment Considering the Manufacturing Quality Assurance Process"

_jne, doi:10.3390/jne4020030_

Round 1

Reviewer 1 Report

1) Interesting work but reason behind selection of dry transformer and reactor pressurizer, with respect to safety function of the entire plant is not clear to this reviewer.

2) Citation of wind turbines, military vehicle and other peripheral systems can be questioned. These systems to not hold the same safety-in-design, safety-in-function implication of a NPP. Please explain further. 

3) This reviewer suggests at least a third example, beyond a dry transformer and reactor pressurizer, especially when the extent of the overall "scaling" between the two in the plant's PSA/PRA model is not described. At least this reviewer did not see this.

4) Overall, to this reviewer, 25 references seems like a small number. The literature in Bayesian methods and applications in nuclear, pre-2000 is likely more extensive than cited. What are the limitations of these applications; is it the lack of manufacturing detail that is given as Tables in the paper?

5) Care should be taken to correlated equipment performance and safety that depends on product quality. The descriptors, "good" and "poor" should be defined, and also avoided because it introduces ambiguity to every reader. Please carefully consider this point and thank you in advance. 

6) Are the BBN tools, AgenaRisk, Netica, validated and verified per nuclear related or relevant standardized or benchmarked or referenced problems? Please explain. 

7) Overall interesting but the paper, as written, generates questions because some details, as noted, could be better explained. 

1) Care should be taken to correlated equipment performance and safety that depends on product quality. The descriptors, "good" and "poor" should be defined, and also avoided because it introduces ambiguity to every reader. Please carefully consider this point and thank you in advance. 

2)  Overall interesting but the paper, as written, generates questions because some details, as noted, could be better explained.  

3) Words that introduce incompleteness or ambiguity should be avoided in technical writing: could, should, may be, might be....etc. Please review submitted draft. 

Author Response

1) Interesting work but reason behind selection of dry transformer and reactor pressurizer, with respect to safety function of the entire plant is not clear to this reviewer.

Response:  Our main purpose of this article is to demonstrate the importance of consideration of manufacturing quality on the reliability analysis. The following sentences are added at the end of ‘Introduction’ at page 2 in the revised manuscript to clarify the selection of dry transformer and reactor pressurizer in this study.

“In nuclear power plants many safety class dry transformers are used in reactor building to feed power to the safety class electromechanical and electronic devices. We selected reactor pressurizer in this study as a mechanical equipment, which is a safety class-1 equipment works under reactor pressure boundary. On the other hand, we selected safety class dry transformer as electrical equipment and pressurizer as mechanical equipment. Similarly the procedure can be applied to the nuclear fuel assemblies, reactor pressure vessel, steam generators, sensors, and reactor protection system.”

2) Citation of wind turbines, military vehicle and other peripheral systems can be questioned. These systems to not hold the same safety-in-design, safety-in-function implication of a NPP. Please explain further. 

Response:  Yes, considering the safety, the above mentioned equipment are different and are not brought under strict safety rules during construction and operation.  Citations of the wind turbines, military vehicle and other peripheral system were brought to present that BBN model was applied in the past for number of cases. FMEA was used for the reliability analysis of wind turbine.

3) This reviewer suggests at least a third example, beyond a dry transformer and reactor pressurizer, especially when the extent of the overall "scaling" between the two in the plant's PSA/PRA model is not described. At least this reviewer did not see this.

Response: We wanted to include another example in our study as per reviewer’s comment. But due to the limitation of Free Version of BBN tools it has become every difficult. Moreover, we have to study the manufacturing process and quality plan of the equipment for the target equipment. Usually safety class mechanical and electromechanical equipment have large quality plan with a large number of control points. So it is very difficult to develop another model for third example within short time.

4) Overall, to this reviewer, 25 references seems like a small number. The literature in Bayesian methods and applications in nuclear, pre-2000 is likely more extensive than cited. What are the limitations of these applications; is it the lack of manufacturing detail that is given as Tables in the paper?

Response: We studied most relevant number of paper and included in the list of references. Five new references have been included in the revised manuscript. The limitations of the applications are collecting and analyzing the detailed manufacturing procedures, test method, and standards.

5) Care should be taken to correlated equipment performance and safety that depends on product quality. The descriptors, "good" and "poor" should be defined, and also avoided because it introduces ambiguity to every reader. Please carefully consider this point and thank you in advance. 

Response:  To clarify the ambiguity to the readers, the following sentences are added at the first paragraph of ‘Introduction’ at page 1 in the revised manuscript

“It is mentioning that good quality assurance means a stringent quality assurance system which has a detailed quality program complying international standard and defined manufacturing procedures.  On the other hand, the poor quality assurance means that it does not have standard quality management system.”

6) Are the BBN tools, AgenaRisk, Netica, validated and verified per nuclear related or relevant standardized or benchmarked or referenced problems? Please explain. 

Response: AgenaRisk was employed for reliability analysis. KAERI employed AgenaRisk for Nuclear I&C software reliability analysis.

7) Overall interesting but the paper, as written, generates questions because some details, as noted, could be better explained.

Response: We would consider reviewer’s comments for our future study.

8) Words that introduce incompleteness or ambiguity should be avoided in technical writing: could, should, may be, might be....etc. Please review submitted draft.

Response: The manuscript is revised according to the comment to clear ambiguity.

Reviewer 2 Report

Minor errors:

Page 1 line 44 and page 2 line 47: FEMA to FMEA

Page 2 line 69: safety grad to safety-grade

Taking into account different meaning in IAEA and NRC regulatory approach it would be good to explicitly mention what is used meaning of safety-grade in context of the paper.

In IAEA documents, where term safety grade is mostly used it is written without dash

In text you said that Figure 1 represents basic QA scheme and in title you are referring to QC systems

Line 90, missing )

Figure 2 caption text, something is missing, modeling approach of what?

Line 130: Familiarization of the -> Familiarization with the

Comments:

Table 1 with QC activities for dry storage transformer is comprehensive, subnets for material, control, manufacturing and tests are appropriate, but what was rationale behind five levels of quality? Assignment of quality level to nodes is still probably mostly qualitative. It would be OK to comment on that. What is meaning of Scenario 1?

What is rationale behind used weighting when combining subnets in overall manufacturing node and when producing overall quality factor?

I can understand that we can implement quality factor as proposed (even though it is not proven that relationship is linear/reciprocal), but in probability of failure/reliability type of analysis we mainly depend on failure rate lambda values. That can be from factory testing or from operational experience but then implicitly takes into account some real level of manufacturing, not ideal one that can scaled up by quality factor? So quality factor can be welcome quantification of quality of manufacturing process but it is not clear if we can use it directly in calculation of failure probability.

All said for dry transformer apply to case of pressurizer too, with the addition that it is not clear how to apply obtained quality factor of 0.97 to failure rate of pressurizer. The pressurizer is not safety-related component and it safety impact is, as presented limited to structural integrity. To assess its failure rate to perform other actions associated equipment should be included (heaters, safety valves, PORVs). That should be clearly stated.

The paper is useful in presenting attempt to quantify influence of manufacturing on the quality of some aspects of the component. The worth of applicability in reliability assessment is not so clear and that should be improved or some kind of title adjustment would be needed. At least pressurizer was not treated from failure probability point of view to the same level as was done for dry transformer.

Some additional spell checking could be used and some blanks are missing after Figure captions.

Author Response

1)  Page 1 line 44 and page 2 line 47: FEMA to FMEA

Response: In the revised manuscript, the word ‘FEMA’ is modified to ‘FMEA’

2) Page 2 line 69: safety grad to safety-grade

Response: In the revised manuscript, the word ‘safety grad’ is modified to ‘safety grade’. The dash is omitted according to comment 4.

3) Taking into account different meaning in IAEA and NRC regulatory approach it would be good to explicitly mention what is used meaning of safety-grade in context of the paper.

 Response: The definition of ‘safety grade’ is required to add at the first paragraph of the section ‘2. Modeling approach for manufacturing quality control process’ at page 3 in the revised manuscript.

4) In IAEA documents, where term safety grade is mostly used it is written without dash.

Response: In the revised manuscript, the word ‘safety grade’ is used throughout the manuscript instead of ‘safety-grade’ according to the comment.

5) In text you said that Figure 1 represents basic QA scheme and in title you are referring to QC systems

Response: The tittle of the Figure 1 is modified in the revised manuscript from ‘Figure1. Basic structure of QC system of a typical NPP safety class equipment’ to ‘Figure1. Basic structure of QA system of a typical NPP safety class equipment’

6) Line 90, missing)

Response: The comment is not understood well.

7) Figure 2 caption text, something is missing, modeling approach of what?

Response: The tittle of the Figure 2 is modified in the revised manuscript from ‘Figure 2. Modeling approach’ to ‘Figure 2. Modeling approach in this study’.

8) Line 130: Familiarization of the -> Familiarization with the

Response: In the revised manuscript, the phrase ‘Familiarization of the’ is modified to ‘Familiarization with the’.

Comments

C1. Table 1 with QC activities for dry storage transformer is comprehensive, subnets for material, control, manufacturing and tests are appropriate, but what was rationale behind five levels of quality? Assignment of quality level to nodes is still probably mostly qualitative. It would be OK to comment on that. What is meaning of Scenario 1?

Response: We thank the reviewer for very constructive comment. The AgenaRisk allows modeling with 3 or five level of rank nodes. We prefer to five levels (very low, Low, Medium, high, Very high) to 3 level( low, medium, high) to reflect better assessment of manufacturing quality. Although the quality assessment for the nodes are qualitative but it gives the opportunity to assess at the root level of manufacturing. Equipment failure and endurance directly related to the manufacturing quality of the equipment.

C2. What is rationale behind used weighting when combining subnets in overall manufacturing node and when producing overall quality factor?

 Response: Actually, it depends on type of equipment and type of quality activities. The recognized that quality activities have different impact on ensuring the product quality. For instance, check of readiness, visual inspection, NDT, or hydraulic test have different impact manufacturing quality. So considered weighting factors for the model. 

C3: I can understand that we can implement quality factor as proposed (even though it is not proven that relationship is linear/reciprocal), but in probability of failure/reliability type of analysis we mainly depend on failure rate lambda values. That can be from factory testing or from operational experience but then implicitly takes into account some real level of manufacturing, not ideal one that can scaled up by quality factor? So quality factor can be welcome quantification of quality of manufacturing process but it is not clear if we can use it directly in calculation of failure probability.

Response: Thanks to the reviewer again for a nice question. Exactly, in reliability analysis, failure rate lambda is used and the value of lambda is estimated from factory testing (FT), operational, or expert judgment (if FT and operational experience data is insufficient). Equipment quality has influence on the reliability of equipment. The quality factor can be estimated and can be used to modify the lambda value. Article 3.4.2 of MIL-HDBK-217 introduced a quality factor  πQ in reliability analysis.

C4. All said for dry transformer apply to case of pressurizer too, with the addition that it is not clear how to apply obtained quality factor of 0.97 to failure rate of pressurizer. The pressurizer is not safety-related component and it safety impact is, as presented limited to structural integrity. To assess its failure rate to perform other actions associated equipment should be included (heaters, safety valves, PORVs). That should be clearly stated.

Response: The following sentences are added at page 13 in the revised manuscript in related to the reactor pressurizer.

“It is mentioning that the NPP reactor pressurizer belongs to the safety class equipment. Failure of pressurizer (e.g. leakage, fracture of vessel, joints or valve, etc.) will introduce LOCA or be an initiating event for small or medium size LOCA (Loss of coolant accident). The quality factor can be used to modify the pressurizer failure rate.”

C5. The paper is useful in presenting attempt to quantify influence of manufacturing on the quality of some aspects of the component. The worth of applicability in reliability assessment is not so clear and that should be improved or some kind of title adjustment would be needed. At least pressurizer was not treated from failure probability point of view to the same level as was done for dry transformer.

Response: Reliability assessment of NPP safety class equipment considering the manufacturing quality assurance process: Case study with Bayesian Belief Network.

Comments on the Quality of English Language

Some additional spell checking could be used and some blanks are missing after Figure captions.

Response: The spellings are checked and blanks are added after Figure captions throughout the manuscript.

Reviewer 3 Report

This paper is very relevant. The new construction of nuclear power plants in developing countries presents a particular nuclear safety challenge. So, the quality control issues in the manufacture of equipment are very important. The issue of the reliability assessment of NPP safety class equipment considering the manufacturing quality assurance process is studied and demonstrated in this paper. The authors propose the so-named “Bayesian belief network” for quality assessment of safety class equipment of NPPs with visible examples.

I especially note the composition of the authors. Two authors from Bangladesh, for whom it is especially important to achieve a high level of reliability and safety for commissioning of first NPP. And one author is from the Republic of Korea, which has both positive and negative experience (in the past) in quality control in the manufacture of NPP equipment.

I recommend for publication without changes.

Author Response

This paper is very relevant. The new construction of nuclear power plants in developing countries presents a particular nuclear safety challenge. So, the quality control issues in the manufacture of equipment are very important. The issue of the reliability assessment of NPP safety class equipment considering the manufacturing quality assurance process is studied and demonstrated in this paper. The authors propose the so-named “Bayesian belief network” for quality assessment of safety class equipment of NPPs with visible examples.

 I especially note the composition of the authors. Two authors from Bangladesh, for whom it is especially important to achieve a high level of reliability and safety for commissioning of first NPP. And one author is from the Republic of Korea, which has both positive and negative experience (in the past) in quality control in the manufacture of NPP equipment.

I recommend for publication without changes.

Response: We thank the reviewer for his constructive and insightful comments. We would like to consider his comments in our future study regarding how to use quality factor in modifying/correcting the mechanical equipment failure.